# Interferon-β Overexpression in Adipose Tissue-Derived Stem Cells Induces HepG2 and Macrophage Cell Death in Liver Tumor Organoids via Induction of TNF-Related Apoptosis-Inducing Ligand Expression

**DOI:** 10.3390/ijms25021325

**Published:** 2024-01-22

**Authors:** Yongdae Yoon, Chang Wan Kim, Moon Young Kim, Soon Koo Baik, Pil Young Jung, Young Woo Eom

**Affiliations:** 1Regeneration Medicine Research Center, Yonsei University Wonju College of Medicine, Wonju 26426, Republic of Korea; yongdae0611@naver.com (Y.Y.); drkimmy@yonsei.ac.kr (M.Y.K.); baiksk@yonsei.ac.kr (S.K.B.); 2Department of Surgery, Yonsei University Wonju College of Medicine, Wonju 26426, Republic of Korea; asparag@naver.com; 3Department of Internal Medicine, Yonsei University Wonju College of Medicine, Wonju 26426, Republic of Korea

**Keywords:** liver tumor organoid, apoptosis, TRAIL, macrophage

## Abstract

Liver tumor organoids derived from liver tumor tissues and pluripotent stem cells are used for liver tumor research but have several challenges in primary cell isolation and stem cell differentiation. Here, we investigated the potential of HepG2-based liver tumor organoids for screening anticancer drugs by evaluating their responsiveness to IFN-β produced by mesenchymal stem cells (MSCs). Liver tumor organoids were prepared in three days on Matrigel using HepG2, primary liver sinusoidal epithelial cells (LSECs), LX-2 human hepatic stellate cells, and THP-1-derived macrophages at a ratio of 4:4:1:1, with 10^5^ total cells. Hepatocyte-related and M2 macrophage-associated genes increased in liver tumor organoids. IFN-β treatment decreased the viability of liver tumor organoids and increased M1 macrophage marker expression (i.e., TNF-α and iNOS) and TRAIL. TRAIL expression was increased in all four cell types exposed to IFN-β, but cell death was only observed in HepG2 cells and macrophages. Further, MSCs overexpressing IFN-β (ASC-IFN-β) also expressed TRAIL, contributing to the reduced viability of liver tumor organoids. In summary, IFN-β or ASC-IFN-β can induce TRAIL-dependent HepG2 and macrophage cell death in HepG2-based liver tumor organoids, highlighting these liver tumor organoids as suitable for anticancer drug screening and mechanistic studies.

## 1. Introduction

Organoids are self-organizing constructs that mimic the in vivo architecture and multi-lineage differentiation of their original tissue [1,2,3]. They can thus be used for modeling diseases like cancer, drug screening, and regenerative medicine research [4,5,6,7,8,9,10,11,12]. In 2009, gut organoids were first developed by Sato et al., who used single leucine-rich repeat-containing G protein-coupled receptor 5 (Lgr5)+ intestinal stem cells [3]. Since then, organoids capable of mimicking various tissues and organs have been established [13]. Liver organoids were first developed by Huch et al. using mouse Lgr5+ liver stem cells [14]; further, Takebe et al. generated liver organoids from human iPSCs combined with endothelial and mesenchymal cells [12]. In addition, Leite et al. and Yoon et al. generated liver organoids from human cell lines [7,8].

Tumor organoids can be used to analyze and recapitulate the genetic features of a tumor, enabling tumor-specific drug screening, and facilitating personalized treatment approaches. Broutier et al. reported that tumor organoids derived from patients with primary liver cancer (PLC) possessed the histological architecture, expression profile, and genomic landscape of the original tumor, with xenografts of PLC-derived organoids exhibiting tumorigenic potential, histological features, and metastatic properties [9]. In addition, Nuciforo et al. showed that tumor organoids obtained from needle biopsies of patients with hepatocellular carcinoma (HCC) of different etiology and tumor stage retained the morphology and tumor marker expression aa well as the genetic heterogeneity of the original tumors. Interestingly, the success rate of tumor organoid generation based on the number of biopsies or patients was ~26 or 33%, respectively [15], which is significantly lower than the reported success rate for pancreatic cancer (75–83%) [16] and colorectal cancer (90%) [17]. The authors suggested that the lower success rate with HCC specimens may be attributed to the fact that hepatocytes, the cells of origin for HCCs, do not possess the epithelial stem cell characteristics needed for their propagation in the organoid culture system [15,18,19]. Further, Wang et al. observed that non-parenchymal cells, such as fibroblasts and endothelial cells, are important in forming tumor organoids and maintaining organoid viability [20].

Recently, we prepared liver organoids using Huh-7 cells/primary liver sinusoidal epithelial cells (LSECs)/LX-2 human hepatic stellate cells/macrophages differentiated from THP-1, and reported the importance of LSECs in organoid formation [8]. Several groups have also used human umbilical vein endothelial cells (HUVECs) as endothelial cells to produce liver organoids [12,20]. However, in our study, HUVECs did not induce self-organization [8]. Further, the TGF-β type I receptor inhibitor A83-01 suppressed organoid formation by Huh-7/primary LSECs/LX-2/macrophages, while TGF-β could generate organoids with Huh-7/LX-2/macrophages in the absence LSECs. The liver organoids formed using Huh-7/primary LSECs/LX-2/macrophages could form fibrotic liver organoids exhibiting detoxification, inflammation, loss of endothelial cell function, and fibrosis in response to thioacetamide, ethanol, and acetaminophen [8]. Taken together, these results suggest that the activity of constituent cells is crucial for liver organoid formation and that liver organoids composed of liver cancer cells and non-parenchymal cells are more suitable for mimicking the liver cancer microenvironment than those composed of liver cancer cells alone.

Self-renewal, multipotent differentiation, secretion of various trophic factors, and migration to damaged areas are important features of mesenchymal stem cells (MSCs) in regenerative medicine, allowing for the regulation of immune cell activity and protecting cells to facilitate the regeneration of damaged tissue [21]. MSCs can migrate to tumors, which are considered non-healing wounds, and promote or inhibit cancer cell growth through several mechanisms [22,23]. Therefore, MSCs have been used as vehicles for loading anticancer drugs, including suicide genes, to locally increase the expression of target genes, thus minimizing the side effects of systemic drug administration and increasing treatment efficacy [24,25]. Adipose tissue-derived stem cells (ASCs) overexpressing either IFN-β or TRAIL (ASC-IFN-β or ASC-TRAIL, respectively) have been reported to inhibit tumor growth in a xenograft model [26,27]. However, the exact therapeutic mechanisms involved in MSC-mediated inhibition of tumor growth in animal models are unclear. Tumor organoids have thus been proposed as a model to evaluate the therapeutic mechanisms and efficacy of MSCs. In this study, we evaluated whether liver organoids prepared using HepG2/primary LSEC/LX-2/macrophages can be used as a liver tumor organoid model for anticancer treatment screens, including that of genetically modified MSCs, by evaluating their responsiveness to IFN-β.

## 2. Results

### 2.1. Generation of Liver Tumor Organoids Using Multiple Cell Types

We have previously demonstrated that liver organoids can be prepared from Huh7/primary LSEC/LX-2/THP-1-derived macrophages to model liver fibrosis [8]. In this study, we analyzed whether these cell line-based liver organoids can be used as an IFN-β-responsive liver tumor organoid model. Using HepG2 cells instead of Huh7, HepG2 cells/primary LSECs/LX-2/THP-1-derived macrophages were cultured on Matrigel to generate liver tumor organoids on day 3; the effect of IFN-β on each cell type in the liver tumor organoids was then analyzed (Figure 1A). HepG2/primary LSEC/LX-2/THP-1-derived macrophages self-organized on Matrigel to form spherical liver tumor organoids on day 3 (Figure 1B). Treatment with the TGF-β type I receptor inhibitor, A83-01, prevented self-organization of HepG2/primary LSECs/LX-2/THP-1-derived macrophages (Figure 1C). Compared with the gene expression of 2D-cultured HepG2/primary LSEC/LX-2/THP-1-derived macrophages, liver tumor organoids showed increased expression of hepatocyte-related genes, increased expression of M1 or M2 markers (IL-1β and TNF-α or IL-10, TGF-β, and CD206, respectively), and significantly increased expression of genes related to fibrosis and endothelial cell function (Figure 1D). These results suggest that the multiple cells constituting liver tumor organoids have retained their functional characteristics. In particular, macrophages express IL-10, TGF-β, and CD206, suggesting that they may act as cancer-associated M2 macrophages in liver cancer organoids.

### 2.2. Effect of IFN-β on Liver Tumor Organoids

Previously, we found that adipose tissue-derived MSCs (ASCs) suppress the growth of hepatocellular carcinoma cells via IFN-β and TRAIL expression [28]. Therefore, in liver tumor organoids, we investigated whether IFN-β could regulate the growth of HepG2 cells and affect the activity of other cells. Treatment of liver tumor organoids with IFN-β for 1 day did not change the spherical organoid shape (Figure 2A); however, organoid viability decreased in an IFN-β dose-dependent manner (Figure 2B). IFN-β induces TRAIL expression in various cell types, and TRAIL induces apoptosis [29,30]. When the liver tumor organoids were treated with TRAIL, their viability decreased (Figure 2C). These results suggest that the viability of liver tumor organoids may be reduced by IFN-β-induced TRAIL expression. Next, to understand how the individual cells that comprise the liver tumor organoids are affected by IFN-β, we analyzed the representative gene expression for each cell type. In liver tumor organoids, IFN-β decreased the expression of albumin (ALB), while upregulating that of AXIN2, CYP3A4, and UGT2B7. Among macrophage-related genes, the expression of TNF-α and iNOS, which are important in inflammatory responses, was increased; interestingly, the expression of TRAIL was markedly increased. IFN-β also increased the expression of COL5A3 and LAMA1 in hepatic stellate cells and of CD31 in LSECs (Figure 2D). These results suggest that IFN-β can induce damage to HepG2 cells and increase the activity of inflammatory M1 macrophages, with minor effects on hepatic stellate cells and LSECs.

### 2.3. TRAIL Expression Induced by IFN-β in 2D-Cultured Cells

The concentration and duration of IFN-β treatment were determined in 2D-cultured HepG2, primary LSECs, LX-2, and THP-1-derived macrophages (Appendix A). To minimize excessive cytotoxicity, we selected 1000 U/mL of IFN-β for 1 day as the treatment conditions. Since IFN-β induces TRAIL expression in various cell types, including macrophages and fibroblasts [31,32], we analyzed TRAIL expression in HepG2, LSECs, LX-2, and THP-1-derived macrophages after IFN-β treatment. Interestingly, IFN-β increased TRAIL expression by approximately 2–9-fold in the four cell types (Figure 3A). Although IFN-β-induced TRAIL expression was observed in all four cell types, reduced cell viability was only observed in HepG2 cells and macrophages (Figure 3B). These results suggest that IFN-β selectively affects the viability of HepG2 cells and macrophages, even though TRAIL expression is induced in all four cell types.

### 2.4. Cell Death Induction by IFN-β in Liver Tumor Organoids

As IFN-β reduced the viability of HepG2 cells and macrophages in a 2D culture, we investigated whether IFN-β would exert a similar effect on these in liver tumor organoids. HepG2 cells and macrophages were distinguished from other cells by immunofluorescence staining using antibodies against CK8 and EMR1, respectively, while dead cells were determined as cleaved caspase 3 (CC3)-positive. IFN-β induced apoptosis in 31.2% of HepG2 cells and 44.9% of macrophages in the liver tumor organoids (Figure 4), while little apoptosis was observed in LX-2 and LSECs. Treatment with inhibitors of JAK, an important component of IFN-β signaling, restored cell viability in liver tumor organoids treated with IFN-β (Figure 5A). Further, TRAIL and CC3 expression in liver tumor organoids was significantly reduced by JAK inhibitors (Figure 5B). These results suggest that in liver tumor organoids, IFN-β can induce the apoptosis of HepG2 cells and macrophages by upregulating TRAIL expression.

### 2.5. Cell Death Induced by ASC-IFN-β in Liver Tumor Organoids

In a previous study, we observed that ASCs overexpressing IFN-β could effectively inhibit tumor proliferation in xenograft models [26,27]. ASCs were transformed with pCMV3-IFN-β, and after 1 day, IFN-β and TRAIL expression was identified by immunoblotting. We also analyzed whether ASCs overexpressing IFN-β (ASC-IFN-β) could regulate the viability of liver tumor organoids. In genetically modified ASCs with pCMV3-IFN-β (ASC-IFN-β), IFN-β expression was increased by approximately 4.4-fold, while TRAIL expression was increased by approximately 13.4-fold compared to that in the controls (Figure 6A). When these ASC-IFN-β were added to liver tumor organoids, the viability of liver tumor organoids was significantly decreased (Figure 6B) and the frequency of dead cells was significantly increased (Figure 6C). These results suggest that ASC-IFN-β can reduce the survival rate of liver tumor organoids owing to IFN-β and TRAIL overexpression.

## 3. Discussion

In this study, we confirmed that HepG2 cells/LSECs/LX-2 cells/macrophages differentiated from THP-1 cells could self-organize to generate liver tumor organoids. IFN-β decreased the viability of liver tumor organoids by inducing the cell death of HepG2 and macrophages. The observed decrease in viability was attributed to IFN-β-induced TRAIL expression in HepG2 cells, LSECs, LX-2 cells, and macrophages. Furthermore, ASCs overexpressing IFN-β (ASC-IFN-β) also expressed TRAIL, contributing to the suppressive effect. Overall, our results suggest that cell-line-based liver tumor organoids can be used to evaluate the applicability of genetically modified MSCs for anticancer therapy.

As IFNs exert pleiotropic effects including cell proliferation and immune system regulation [33,34], they have been employed for the treatment of several diseases, including cancer [29]. IFNs exhibit growth-inhibitory and apoptotic effects in various cancers through TRAIL induction [30,35,36,37,38]. Interestingly, even though the TRAIL expression level in LX-2 cells was lower than that in the other cells, IFN-β induced TRAIL expression in all the HepG2, LX-2, LSECs, and macrophages. However, IFN-β induced cell death only in HepG2 cells and macrophages. Notably, TRAIL is known to selectively induce apoptosis in tumor cells without harming normal cells [39,40]. TRAIL binds to TRAIL death receptors, DR4 and DR5, on tumor cells, leading to tumor cell apoptosis, but does not induce apoptosis in normal cells because they express TRAIL decoy receptors, DcR1, DcR2, and osteoprotegerin (OPG) [41,42,43]. Therefore, TRAIL has attracted attention for selectively eliminating tumor cells without harming normal cells [39,40]. Among the cells used to generate liver tumor organoids in this study, HepG2 cells and macrophages were derived from hepatocellular carcinoma and acute monocytic leukemia cells, respectively, LSECs were primary cells, and the LX-2 cell line was created by immortalizing primary hepatic stellate cells. Thus, IFN-β-induced TRAIL caused apoptosis only in the HepG2 cancer cells and macrophages. Notably, Nanogold-TRAIL complexes are known to increase cytotoxicity in M2 rather than M1 macrophages [44], with TRAIL promoting M2-to-M1 phenotype shifts [45]. In our study, M2 markers IL-10, TGF-β, and CD206 were increased in the liver tumor organoids compared to those in the 2D-cultured cells. However, IFN-β increased the M1 markers TNF-α and iNOS expression in liver tumor organoids. These results suggest that IFN-β treatment of liver tumor organoids may promote an M2-to-M1 phenotype shift as well as macrophage death. In summary, IFN-β may contribute to tumor therapy via TRAIL expression by inducing the death of cancer cells and M2 macrophages as well as by increasing the activity of M1 macrophages.

Chemotherapy is very helpful in suppressing cancer cells, but it also has side effects that damage normal cells. Since these side effects can be reduced to some extent by administering anticancer agents locally rather than systemically, MSCs can be utilized to induce localized expression of anticancer agents. Therefore, preclinical and clinical studies using MSCs or genetically engineered MSCs to treat cancer are actively underway [46,47,48]. In particular, TRAIL is being used as a therapeutic drug to minimize the side effects of cancer treatment because it can selectively kill cancer cells without harming normal cells [39,40]. As MSCs have low gene transfer efficiency, gene-delivery techniques using viral vectors rather than non-viral vectors have been applied [24,25,49,50]. However, viral vectors have the disadvantages of potential immunogenicity and insertional mutagenesis [51,52]. In this study, we prepared ASCs expressing IFN-β (ASC-IFN-β) using a plasmid, that is, a non-viral vector, and observed that the resulting ASC-IFN-β could induce apoptosis in HepG2 cells and macrophages. Theoretically, even if only some MSCs express IFN-β, ASC-IFN-β is expected to compensate for the lower transfection efficiency of MSCs in expressing TRAIL and increasing local TRAIL expression compared to that with ASC-TRAIL because IFN-β can induce TRAIL expression in the surrounding MSCs or tumor cells. Indeed, in a xenograft model, ASC-IFN-β could inhibit tumor growth more effectively than ASC-TRAIL [26,27]. Therefore, ASCs expressing IFN-β introduced via a plasmid have greater clinical potential than those transduced with a viral vector. Taken together, increased M1 macrophage markers TNF-α and iNOS and apoptosis induction in HepG2 cells and macrophages from IFN-β-exposed liver tumor organoids suggest the selective tumor therapeutic potential of IFN-β. Further, as ASC-IFN-β enables local expression of IFN-β and TRAIL, it could be used as a tumor therapy to minimize the side effects of systemic administration. Since MSCs can migrate to the tumor site and ASC-IFN-β can locally express IFN-β and TRAIL at high concentrations, the clinical applicability of ASC-IFN-β is greater than that of IFN-β and TRAIL administration. However, further studies are needed to evaluate the local expression of IFN-β and TRAIL in animal models infused with ASC-IFN-β as well as to determine whether ASC-IFN-β has fewer side effects than those induced by the systemic administration of IFN-β.

Preparing liver organoids composed of multiple cell types rather than hepatocytes alone (or together with biliary epithelial cells), is critical to recapitulating liver physiology. Ultimately, preparing liver tumor organoids using patient-derived HCC cells could enable patient-specific drug screening, which could lead to fewer adverse effects and increased efficiency of liver cancer treatment. Currently, we have confirmed that liver tumor organoids prepared using Huh-7 instead of HepG2 exhibit similar responsiveness to IFN-β (Appendix A). Further, we are preparing liver tumor organoids using other liver cancer cell lines. We are also preparing liver organoids using primary cells isolated from rat or mouse livers. In the future, we plan to prepare liver tumor organoids using primary cells isolated from liver cancer patients. The goal is to employ these patient-derived organoids in order to study therapeutic mechanisms of treatment and per-form personalized drug screening, with the goal of minimizing the side effects of cancer drugs while maximizing therapeutic efficacy.

## 4. Materials and Methods

### 4.1. Cell Culture

HepG2 and THP-1 cells were purchased from the Korea Cell Line Bank (Seoul, Republic of Korea), LX-2 cells were purchased from Millipore (Burlington, MA, USA), and primary LSECs were purchased from ScienCell (Carlsbad, CA, USA). HepG2 and THP-1 cells were cultured in Dulbecco’s Modified Eagle Medium (DMEM; Gibco BRL, Rockville, MD, USA) and Roswell Park Memorial Institute Medium 1640 (RPMI, Gibco BRL), respectively, with 10% fetal bovine serum (FBS; Gibco BRL) and penicillin/streptomycin (Gibco BRL). Prior to liver tumor organoid generation, THP-1 monocytes were differentiated into macrophages using phorbol 12-myristate-13-acetate (100 nM, Sigma-Aldrich, St. Louis, MO, USA) for two days. LX-2 cells were maintained in DMEM (Gibco BRL) with 3% FBS and penicillin/streptomycin. Endothelial cell medium (ECM; ScienCell) was used for maintaining primary LSECs (ScienCell). LSECs at passage four were used in this study because LSECs beyond five passages show increased α-SMA expression and decreased CD31 expression [8]. ASCs were isolated from three healthy donors (24–38 years of age) with their written informed consent through an elective liposuction procedure under anesthesia at the Wonju Severance Christian Hospital (Wonju, Republic of Korea). ASCs were isolated using a modified protocol described by Zuk et al. [53] and subcultured in DMEM (Gibco BRL) with 10% FBS and penicillin/streptomycin. ASCs with less than five passages were used in this study. ASCs expressing IFN-β were prepared by transfection using pCMV3-IFN-β plasmid DNA (SinoBiological, Wayne, PA, USA) and X-tremeGENE HP DNA transfection reagent (Roche, Basel, Switzerland). All cells were incubated at 37 °C with 5% CO_2_.

### 4.2. Liver Tumor Organoid Generation

A mixture of equal parts DMEM and ECM (DMEM/ECM; 1:1) was used to culture liver tumor organoids. Matrigel (Corning Life Sciences, Corning, NY, USA) was mixed with DMEM/ECM at a 1:1 ratio, then 50 µL per well was added to a 96-well plate and gelled in a cell culture incubator for at least 10 min. To prepare liver tumor organoids, HepG2 cells, LSECs, macrophages differentiated from THP-1, and LX-2 cells were mixed at a ratio of 4:4:1:1. A total of 1 × 10^5^ cells were seeded in Matrigel-coated wells; the medium was replaced with fresh medium on days 2 and 3. For 2D culture, HepG2 cells, LSECs, macrophages, and LX-2 cells were cultured in 96-well plates without a Matrigel coating. To determine whether TGF-β plays an important role in the process of liver tumor organoid generation, the TGF-β type 1 receptor inhibitor A83-01 was added at a concentration of 500 nM on day 0. IFN-β (1000 or 10,000 U/mL), TRAIL (10 or 50 ng/mL), ASC-IFN-β (1 × 10^4^ cells), and JAK inhibitors (10 µM each of JAK1 and JAK2 inhibitors) were added on day 3.

### 4.3. Liver Tumor Organoid Viability Assay

Liver tumor organoid viability was analyzed using the WST-1 and the live/dead cell assay. After treating the liver tumor organoids with IFN-β or TRAIL for 1 day, liver tumor organoid viability was measured using the WST-1 Assay Kit (ab65473; Abcam, Cambridge, UK) according to the manufacturer’s instructions. Briefly, 15 µL of WST-1 reagent was added to the liver tumor organoids and incubated for 2 h. The optical density at 440 nm was then measured using a microtiter plate reader (Molecular Devices; San Jose, CA, USA). Further, cell death in liver tumor organoids treated with ASC-IFN-β was analyzed using a Live/Dead viability kit (L3224; Invitrogen, Carlsbad, CA, USA). Briefly, 5 µL of calcein-AM and 20 µL of ethidium homodimer-1 were diluted in 10 mL DPBS, and 150 µL was added to liver tumor organoids from which the medium was removed and reacted for 30 min; the cells were then observed under a fluorescence microscope (Eclipse TS2R, Nikon, Japan). The viability of 2D-cultured cells was also analyzed using the WST-1 Assay Kit (ab65473; Abcam).

### 4.4. Immunoblotting

Liver tumor organoids were lysed using sample buffer [62.5 mM Tris-HCl, pH 6.8, 34.7 mM sodium dodecyl sulfate (SDS), 5% β-mercaptoethanol, and 10% glycerol], sonicated, boiled for 5 min, subjected to SDS-polyacrylamide gel electrophoresis, and transferred onto a polyvinylidene difluoride membrane (Millipore). Sonication was performed using a VCX 500 sonicator (Sonics, Newtown, CT, USA) with 20% amplitude for 4 s in a 1 s-on/1 s-off mode. The membrane was blocked with 5% skim milk in Tris-HCl-buffered saline containing 0.05% Tween 20 (TBST) for 30 min and incubated overnight at 4 °C with primary antibodies against TRAIL (AF375; R&D Systems, Minneapolis, MA, USA), cleaved caspase-3 (CC3, MAB835; R&D Systems), caspase-3 (pro-C3, MAB707; R&D Systems), and GAPDH (sc47724; Santa Cruz Biotechnology, Santa Cruz, CA, USA). The membrane was washed thrice for 5 min with TBST and incubated with horseradish peroxidase-conjugated secondary antibodies (7074S and 7076S; Cell signaling Technology Danvers, MA, USA, or AP106P; Millipore) for 1 h. After washing thrice with TBST, the bands were visualized using EZ-Western Lumi Pico or Femto (Dogen, Seoul, Republic of Korea) and recorded using a ChemiDoc XRS+ System with Image Lab^TM^ software Version 5.2.1 (Bio-Rad Laboratories, Hercules, CA, USA).

### 4.5. Quantitative Polymerase Chain Reaction (qPCR)

Three liver tumor organoids were pooled into one, and total RNA was isolated using TRIzol reagent (Gibco BRL). cDNA was synthesized from total RNA (1 µg) using a Verso cDNA Synthesis Kit (ThermoFisher Scientific, Waltham, MA, USA). The cDNA was mixed with a primer and power SYBR Green PCR Master Mix (Applied Biosystems, Dublin, Ireland) and amplified using a QuantStudio 6 Flex Real-time PCR system (ThermoFisher Scientific). The primer sequences used in this study are indicated in Appendix A. Fold changes in mRNA expression were determined using the 2^−(△△Ct)^ method.

### 4.6. Immunofluorescence Staining

Liver tumor organoids were fixed in 4% paraformaldehyde (Tech and Innovation, Chuncheon, Republic of Korea) for 1 h and incubated overnight at 4 °C in 30% sucrose phosphate buffer. Thereafter, the liver organoids were embedded in FSC-22 frozen section medium (Leica Biosystems, Wetzlar, Germany) and cut into 10 µm-thick sections using a cryostat at −20 °C. The sections were then permeabilized using 1% Triton X100 in phosphate-buffered saline (PBS) for 10 min, and blocked using blocking buffer (5% bovine serum albumin in PBS containing 0.2% Triton X100) for 30 min. The sections were then incubated overnight at 4 °C with primary antibodies against CK8 (sc-8020; Santa Cruz Biotechnology), EMR1 (sc-377009; Santa Cruz Biotechnology), and cleaved caspase-3 (CC3, MAB835; R&D Systems) in blocking buffer. For fluorescence labeling, the sections were incubated with Alexa Fluor 488- or Alexa Fluor Plus 647-conjugated secondary antibody (1:100; Invitrogen, Carlsbad, CA, USA) for 1 h at 25 °C. Nuclear staining was conducted using DAPI (4′,6-diamidino-2-phenylindole; Sigma-Aldrich, St. Louis, MO, USA), and samples were mounted using DPX Mounting Medium (Electron Microscopy Sciences, Hatfield, PA, USA). The sections were then observed and imaged under a confocal microscope (LSM800; Zeiss, Oberkochen, Germany).

### 4.7. Statistical Analyses

All statistical analyses were performed using one-way analysis of variance, followed by Tukey’s post hoc tests. A two-tailed Student’s *t*-test was used to evaluate the differences between the two groups. Data are presented as the mean ± standard deviation (SD). All *p* values < 0.05 were considered statistically significant.

## 5. Conclusions

HepG2 cells/LSECs/LX-2 cells/macrophages differentiated from THP-1 could self-organize into liver tumor organoids. In these liver tumor organoids, macrophages expressed the M2 markers IL-10, TGF-β, and CD206. Treatment with IFN-β reduced the viability of liver tumor organoids by inducing the apoptosis of HepG2 cells and macrophages, in addition to upregulating the expression of M1 markers TNF-α and iNOS in macrophages. The decrease in organoid viability was attributed to IFN-β-induced TRAIL expression in HepG2 cells, LSECs, LX-2 cells, and macrophages. Further, IFN-β was shown to induce the M2-to-M1 transition of macrophages. ASCs overexpressing IFN-β (ASC-IFN-β) also expressed TRAIL and suppressed liver tumor organoid viability. Overall, our results suggest that cell line-based liver tumor organoids can be used to screen anticancer therapies based on IFN-β or ASC-IFN-β and interrogate the underlying mechanisms.

## Figures and Tables

**Figure 1 ijms-25-01325-f001:**
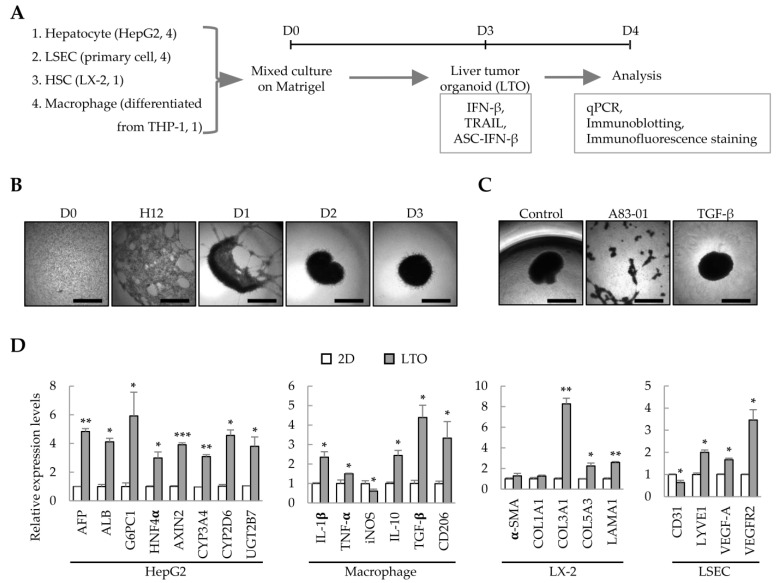
Liver tumor organoids generated using multiple cell types. To generate self-organized liver tumor organoids, HepG2 cells, primary liver sinusoidal endothelial cells (LSECs), LX-2 hepatic stellate cells, and macrophages differentiated from THP-1 cells were seeded together in a Matrigel-coated 96-well plate and cultured for three days. Responsiveness to IFN-β, TRAIL, or ASC-IFN-β in liver tumor organoids was determined on day 4. (**A**) Schematic diagram of the experimental protocol. (**B**) Representative images at the indicated time points (days 0–3). The medium was replaced with fresh medium on days 2 and 3. (**C**) Inhibition of self-assembly of multiple cells by A83-01. A83-01, a TGF-β receptor inhibitor, inhibited the self-assembly of the four cell types into organoids. Scale bar for (**B**,**C**): 1 mm. (**D**) Gene expression profiles in 2D- and 3D-cultured cells. The four cell types were cultured for three days in 96-well plates without or with Matrigel for 2D or 3D cultures, respectively. Target gene expression was normalized using *GAPDH* expression. Relative fold changes in mRNA expression were measured using the 2^−(ΔΔCt)^ method. The results are presented as the mean ± standard deviation (SD) of three replicates. * *p* < 0.05, ** *p* < 0.01, and *** *p* < 0.001. 2D: 2D-cultured cells; LTO: 3D-cultured liver tumor organoids.

**Figure 2 ijms-25-01325-f002:**
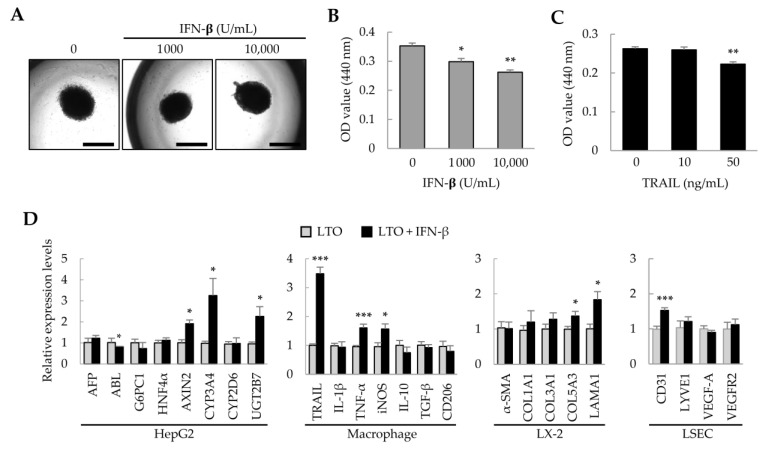
Effect of IFN-β on liver tumor organoids. On day 3, liver tumor organoids were treated with IFN-β for an additional day. (**A**) Representative images after IFN-β treatment. Liver tumor organoids exposed to IFN-β retained their spherical shape. Scale bar: 1 mm. (**B**,**C**) Decreased viability of liver tumor organoids induced by IFN-β or TRAIL. (**D**) Gene expression profiles in IFN-β-treated liver tumor organoids. Target gene expression was normalized to that of *GAPDH*. Relative fold changes in mRNA expression were measured using the 2^−(ΔΔCt)^ method. Data are presented as the mean ± standard deviation (SD) of three replicates. * *p* < 0.05, ** *p* < 0.01, and *** *p* < 0.001. LTO: 3D-cultured liver tumor organoids; LX-2: human hepatic stellate cells; LSEC: primary liver sinusoidal endothelial cells.

**Figure 3 ijms-25-01325-f003:**
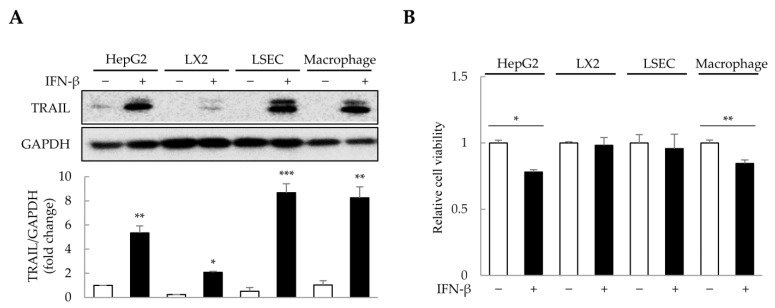
IFN-β-induced TRAIL expression in 2D-cultured cells. To determine the cells that express TRAIL and whose viability would be affected by IFN-β, 2D-cultured HepG2 cells, primary liver sinusoidal endothelial cells (LSECs), LX-2 human hepatic stellate cells, and macrophages were treated with IFN-β for 1 day. TRAIL expression was then analyzed via immunoblotting, and cell viability was determined using the WST-1 assay. (**A**) IFN-β-induced TRAIL expression in the four cell types. The intensity of protein bands was quantified through densitometry using ImageJ (National Institutes of Health, https://imagej.net/ij/index.html, 10 August 2023), and its relative expression was normalized to that of GAPDH. Data are presented as the mean ± SD of three independent experiments. * *p* < 0.05, ** *p* < 0.01, and *** *p* < 0.001. (**B**) Cell viability upon IFN-β treatment in the four cell types. Data are presented as the mean ± SD of three independent experiments. * *p* < 0.05 and ** *p* < 0.01.

**Figure 4 ijms-25-01325-f004:**
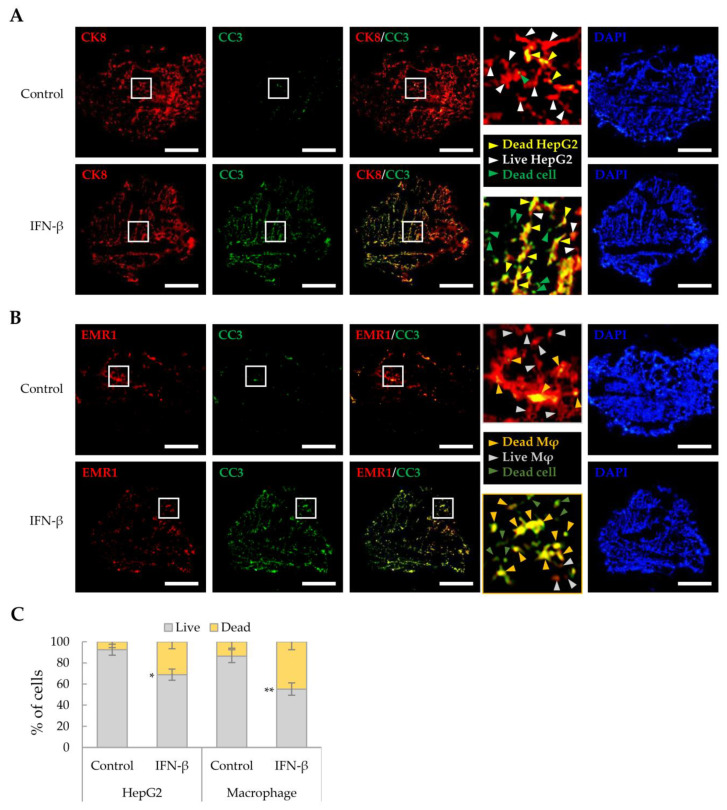
Cell death induced by IFN-β in liver tumor organoids. To label HepG2 cells or macrophages, antibodies against CK8 or EMR1, respectively, were used for staining liver organoid sections obtained on day 4 with and without IFN-β treatment. Dead cells were evaluated for cleaved-caspase 3 (CC3) expression. (**A**) Death of CK8-positive HepG2 cells in liver tumor organoids treated with IFN-β. Green arrowheads indicate dead cells other than HepG2. Yellow arrowheads indicate dead HepG2 cells. White arrowheads indicate live HepG2 cells. Scale bar: 500 µm. (**B**) Cell death of EMR1-positive macrophages in liver tumor organoids treated with IFN-β. Dark green arrowheads indicate dead cells other than macrophages. Dark yellow arrowheads indicate dead macrophages. Gray arrowheads indicate live macrophages. Scale bar: 500 µm. In (**A**,**B**), the white boxes indicate areas that have been enlarged to distinguish the staining of the cells. (**C**) Quantitative analysis of IFN-β-induced cell death in HepG2 cells and macrophages in liver tumor organoids. The number of CC3-positive HepG2 cells and macrophages was counted in three independent experiments, with more than 500 cells counted per experiment. Data are presented as the mean ± SD of three independent experiments. * *p* < 0.05 and ** *p* < 0.01.

**Figure 5 ijms-25-01325-f005:**
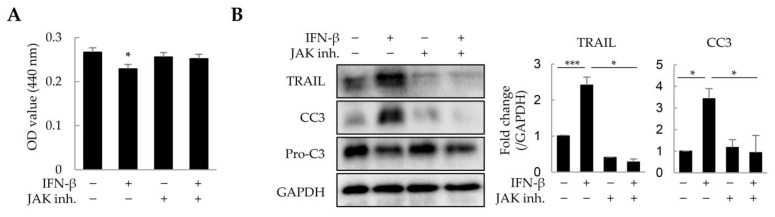
Suppression of IFN-β-induced cell death by JAK inhibitors in liver tumor organoids. Liver tumor organoids were treated with JAK inhibitors (10 µM each of JAK1 and JAK2 inhibitor) for 20 min prior to IFN-β treatment. After 24 h, the viability of organoids was analyzed using the WST-1 assay, and the expression of TRAIL and cleaved-caspase 3 (CC3) was analyzed by immunoblotting. (**A**) Viability of liver tumor organoids treated with IFN-β and/or JAK inhibitors. (**B**) Expression of TRAIL and cleaved-caspase 3 (CC3) in liver tumor organoids treated with IFN-β and/or JAK inhibitors. The results are presented as the mean ± standard deviation (SD) of three replicates. * *p* < 0.05 and *** *p* < 0.001; inh: inhibitor.

**Figure 6 ijms-25-01325-f006:**
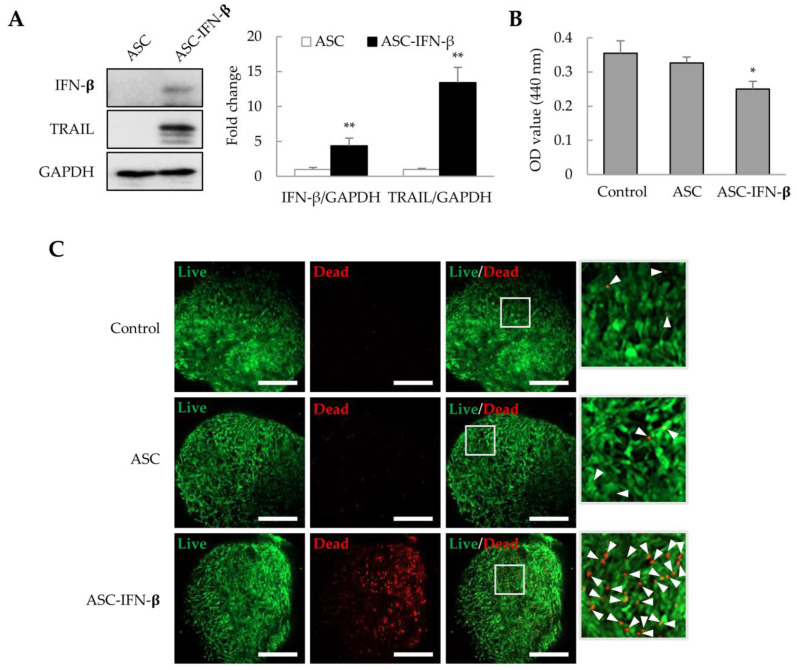
Cell death in liver tumor organoids induced by ASC-IFN-β. Adipose-derived stem cells (ASCs) were transiently transfected with pCMV3-IFN-β. IFN-β and TRAIL expression was determined via immunoblotting after 1 day. ASC-IFN-β was added to liver tumor organoids on day 3, and ASC-IFN-β-induced viability and cell death in liver tumor organoids were examined on day 4. (**A**) IFN-β and TRAIL expression in ASC-IFN-β. The intensity of protein expression was quantified through densitometry using ImageJ (National Institutes of Health, https://imagej.net/ij/index.html, 10 August 2023), and the relative expression was normalized to that of GAPDH. (**B**) Suppression of liver tumor organoid viability by ASC-IFN-β. The viability of organoids was analyzed using the WST-1 assay. (**C**) Cell death induced by ASC-IFN-β in liver tumor organoids. ASC-IFN-β-induced cell death was evaluated using a live (calcein-AM) and dead (ethidium homodimer-1) assay. In (**C**), the white boxes indicate areas that have been enlarged to distinguish the staining of the cells. White arrowheads indicate dead cells. Data shown are representative of three independent experiments. * *p* < 0.05 and ** *p* < 0.01. Scale bar: 500 µm.

## Data Availability

The data supporting the findings of this study are available from the corresponding author upon reasonable request.

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
