# Peer review of "Interferon-β Overexpression in Adipose Tissue-Derived Stem Cells Induces HepG2 and Macrophage Cell Death in Liver Tumor Organoids via Induction of TNF-Related Apoptosis-Inducing Ligand Expression"

_ijms, 2024, doi:10.3390/ijms25021325_

Round 1
Reviewer 1 Report
Comments and Suggestions for Authors
The authors aim to generate organoids that recapitulate liver tumours, bu using a mixture of liver populations (Liver tumor organoids were prepared in three days on Matrigel using HepG2, 20 primary liver sinusoidal epithelial cells (LSECs), LX-2 human hepatic stellate cells, and THP-1-de- 21 rived macrophages in a ratio of 4:4:1:1).
The use of all these populations, instead of only hepatocytes (and maybe biliary epithelial cells) implies a very important step in getting closer to the liver physiology in vitro.
The Introduction briefly and properly summarizes recent contributions to the generation of liver organoids with many cell types.
Results are detailed and the comparison with 2D cell cultures is very meaningful.
Some comments are:
HepG2 is a cell line isolated from a hepatocellular carcinoma of a Causasian male teenager. Considering that your protocol works, are you thinking about using a female hepatocellular carcinoma and/or primary cells isolated from specific patients? Could you comment on that on the Discussion?
You have used Matrigel. We all have observed the batch to batch variability of Matrigel (as well as the difficulties to purchase it). Are you considering any protocol to culture it without Matrigel?
Comments on the Quality of English LanguageEnglish language is easy to understand.
Reviewer 2 Report
Comments and Suggestions for Authors
The research article titled "IFN-β Overexpression in Adipose Tissue-Derived Stem Cells 2 Induces HepG2 and Macrophage Cell Death in Liver Tumor 3 Organoids via Induction of TRAIL Expression" gives an interesting investigation into the prospect of employing liver tumor organoids for drug screening purposes. Nevertheless, there are a few shortcomings and areas where experimental enhancements might augment the quality of the research:
1. Utilizing HepG2 cells as a model for liver tumor organoids could have a few limitations. HepG2 cells, classified as a hepatocellular carcinoma cell line, may not accurately reflect the complex structure of primary liver cancers. Engaging in a thorough analysis of the difficulties and potential implications associated with utilizing HepG2 cells might enhance the article. Furthermore, the use of HuH7 together with HepG2 has the potential to augment the quality of the data.
2. Although the study addresses the activation of TRAIL expression and the resulting cell death, a more comprehensive analysis of the precise processes and pathways implicated in TRAIL-induced apoptosis could improve the understanding of the results.
3. The study should provide a discussion of the practical significance of the findings in a clinical context. What is the connection between the reported impacts of IFN-β and ASC-IFN-β on liver tumor organoids and their possible application in therapeutic techniques or medication development? Including in a discussion about these elements would enhance the scope and impact of the research.
4. Conduct dose-response and time-course experiments in which the concentration of IFN-β changes in order to ascertain whether its impact on organoid viability is dependent on dosage. This study aims to determine the ideal concentration at which TRAIL-dependent cell death can be induced while minimizing excess cytotoxicity.
Comments on the Quality of English Language
no any
Round 2
Reviewer 2 Report
Comments and Suggestions for Authors
I am satisfied with the author's comments and responses.
Comments on the Quality of English LanguageNo any